# Serotonin transporter genotype modulates resting state and predator stress-induced amygdala perfusion in mice in a sex-dependent manner

**Jann F. Kolter[1,2], Markus F. Hildenbrand[3], Sandy Popp[2], Stephan Nauroth[2], Julian Bankmann[1], Lisa Rother[1], Jonas Waider[2], Jürgen Deckert[1], Esther Asan[4], Peter M. Jakob[5], Klaus-Peter Lesch[2], Angelika Schmitt-Böhrer[1] ***

1 Department of Psychiatry, Psychosomatics and Psychotherapy, Center of Mental Health, University of Wuerzburg, Wuerzburg, Germany, 2 Division of Molecular Psychiatry, Department of Psychiatry, Psychosomatics and Psychotherapy, Center of Mental Health, University of Wuerzburg, Wuerzburg, Germany, 3 Department of Magnetic Resonance and X-Ray Imaging, Fraunhofer Development Center X-Ray Technology, Wuerzburg, Germany, 4 Institute of Anatomy and Cell Biology, University of Wuerzburg, Wuerzburg, Germany, 5 Department of Experimental Physics 5, University of Wuerzburg, Wuerzburg, Germany

* Schmitt_A3@ukw.de

**Data Availability Statement:** All relevant data are within the manuscript and its Supporting Information files.

## Abstract

The serotonin transporter (5-HTT) is a key molecule of serotoninergic neurotransmission and target of many anxiolytics and antidepressants. In humans, *5-HTT* gene variants resulting in lower expression levels are associated with behavioral traits of anxiety. Furthermore, functional magnetic resonance imaging (fMRI) studies reported increased cerebral blood flow (CBF) during resting state (RS) and amygdala hyperreactivity. 5-HTT deficient mice as an established animal model for anxiety disorders seem to be well suited for investigating amygdala (re-)activity in an fMRI study. We investigated wildtype (5-HTT+/+), heterozygous (5-HTT+/-), and homozygous 5-HTT-knockout mice (5-HTT-/-) of both sexes in an ultra-high-field 17.6 Tesla magnetic resonance scanner. CBF was measured with continuous arterial spin labeling during RS, stimulation state (SS; with odor of rats as aversive stimulus), and post-stimulation state (PS). Subsequently, *post mortem* c-Fos immunohistochemistry elucidated neural activation on cellular level. The results showed that in reaction to the aversive odor CBF in total brain and amygdala of all mice significantly increased. In male 5-HTT+/+ mice amygdala RS CBF levels were found to be significantly lower than in 5-HTT+/- mice. From RS to SS 5-HTT+/+ amygdala perfusion significantly increased compared to both 5-HTT+/- and 5-HTT-/- mice. Perfusion level changes of male mice correlated with the density of c-Fos-immunoreactive cells in the amygdaloid nuclei. In female mice the perfusion was not modulated by the *5-Htt*-genotype, but by estrous cycle stages. We conclude that amygdala reactivity is modulated by the *5-Htt* genotype in males. In females, gonadal hormones have an impact which might have obscured genotype effects. Furthermore, our results demonstrate experimental support for the tonic model of *5-HTT*LPR function.

**Funding:** This work was funded by the German Research Foundation (DFG; https://www.dfg.de/), grant number 44541416: Collaborative Research Centre/Transregio 58 (CRC-TRR58), subproject A1 to KPL and subproject A5 to KPL and ASB. The funder had no role in study design, data collection and analysis, decision to publish or preparation of the manuscript.

**Competing interests:** The authors have declared that no competing interests exist.

## Introduction

The monoamine serotonin (5-hydroxtryptamine, 5-HT) is one of the key modulators of emotional states and cognitive processing. The 5-HT transporter (5-HTT) responsible for the reuptake of synaptically released 5-HT into the presynaptic neuron is important for the fine-tuning of serotonergic neurotransmission and is a principal target of various antidepressants and anxiolytics [1–3]. In humans, variation of *5-HTT/SLC6A4* gene expression levels are mainly genetically driven by the *5-HTT* gene-linked polymorphic region (*5-HTT*LPR) in the promotor. The short (S)-allele results in lower 5-HTT mRNA and protein levels and is shown to be associated with an increased risk for affective disorders and maladaptive behavioral traits [4–7]. Additionally, a single-nucleotide polymorphism in the long (L) variant affects 5-HTT availability with the $L_G$ variant resulting in nearly equivalent expression levels as the S variant [8].

To further deepen the knowledge of the influence of altered 5-HT neurotransmission on neurodevelopment and behavior the 5-HTT knockout (-/-) mouse model with a targeted disruption of the *5-Htt* gene was generated [9]. 5-HTT-deficient mice exhibit many changes at the neurochemical/5-HT receptor level [10–14], display increased anxiety-related behavior [15,16] and altered stress susceptibility [17,18], and therefore are established as an animal model for anxiety disorders and for *5-Htt* gene-by-environment interaction studies[19–21]. A previous study shows that olfactory perception is unaltered in 5-HTT-deficient mice [22].

Understanding of fear and anxiety disorders requires the elucidation of networks and processes that convert an aversive stimulus into a fear reaction irrespective of whether it is appropriate or not. The anxiety/fear network comprises several directly or indirectly connected regions and nuclei, including the amygdala, prelimbic and infralimbic cortex, bed nucleus of stria terminalis, hippocampus, and periaqueductal grey including the raphe nuclei [for review [23]]. Our research mainly focuses on the amygdala, as it is an important key player in the neurocircuitry of fear, stress, and anxiety disorders [24]. The amygdala, a complex structure consisting of several interconnected nuclei, receives numerous afferents from different brain regions including serotonergic afferents primarily originating in neurons of the dorsal and sparsely originating in neurons of the median raphe nuclei [25–27]. The two main sub-areas of the amygdala are the striatum-like central nucleus of the amygdala (Ce) composed mainly of GABAergic neurons and the cortex-like basolateral amygdala (BLA) with around 80% glutamatergic principal neurons and roughly 20% GABAergic interneurons. The BLA is composed of the lateral (La) and basolateral (BL) nucleus [28,29].

Several behavioral responses important for the survival and reproduction of an organism like feeding, mating and escaping are known to be initiated and driven by odors. The medial nucleus of the amygdala receives olfactory information either by the vomeronasal or the main olfactory system [30–33] Chemosensory information from both pathways is further transferred to the hypothalamus, nucleus accumbens and to the Ce and BLA [34]. Mice that are exposed to rat predator scents exhibit innate defensive behaviors including flight and freezing as well as an increase in stress hormone levels [35–39]. Furthermore, predator odors were shown to evoke an increase in the immediate early gene (IEG) product c-Fos in the BL, Ce and medial nucleus in rodents [40–42]. Increase in c-Fos protein expression occurs in response to direct stimulation of neurons and serves as marker for neuronal activation [43–45].

In human fMRI studies, increased resting cerebral blood flow (CBF) in the amygdala and hippocampus of healthy *5-HTT/SLC6A4* S-allele carriers compared to L-allele carriers was demonstrated first by Canli and coworkers using arterial spin labeling (ASL) [46]. These findings were independently replicated for the amygdala using continuous ASL (CASL), whereas no difference in global CBF intensities was found across the two genotype groups [47]. Controversially to all these findings, Viviani and coworkers did not reveal an association

between the *5-HTT*LPR polymorphism and baseline brain perfusion in a cohort of 183 healthy individuals appying CASL [48].

Applying another technique, the blood oxygen level dependent (BOLD) contrast imaging, healthy individuals with one or two copies of the S allele exhibited greater BOLD signal change in the amygdala in response to the presentation of fearful and angry faces indicating increased neuronal activity in the amygdala of S allele carriers compared to the homozygous L allele group [[49–53] for review: [54,55]]. Results of 99mTc-HMPAO-SPECT scans performed with patients suffering from major depression also pointed to an over(re)activity of the amydalae of the S-allele group relative to the L/L group [56].

Amygdala activity and reactivity to negative stimuli can be investigated in mice, as in humans, with the CASL method using magnetically labelled blood as an intrinsic marker to measure perfusion and signal changes in specific brain regions [57]. In the present study, the CASL approach was applied using an ultra-high-field 17.6 Tesla magnetic resonance imaging (MRI) system to investigate CBF and predator odor-induced perfusion changes in the amygdala and whole brain of male and female mice of different *5-Htt* genotypes. As a marker for perfusion this study uses a perfusion level that indicates the proportion of perfusion relative to the overall signal intensity of each voxel, which represents a much less intereference-prone parameter. This approach extensively excludes any preexisting and constant conditions like subject specific blood supply, which might influence the measurements. Subsequently, quantitative immunohistochemistry for c-Fos was carried out to assess amygdala activation at the cellular level, combining, for the first time, ultra-high-field MRI at 17.6 T and IEG-based cellular activation mapping strategies. We hypothesized that rat odor-induced perfusion level changes and c-Fos-immunoreactivity in the rodent amygdala are differentially affected by *5-Htt* genotype and sex.

## Material and methods

### Ethics statement

The present work complies with current regulations regarding animal experimentation in Germany and the EU (European Communities Council Directive 86/609/EEC). All procedures and protocols have been approved by the committees on the ethics of animal experiments of the University of Würzburg and of the Government of Lower Franconia (license 55.2–2531.01-81/10). Sacrifice was performed under deep isoflurane anesthesia. All efforts were made to minimize suffering.

### Animals

Experimental subjects were 3–6 month-old 5-HTT homozygous knockout (5-HTT-/-), heterozygous knockout (5-HTT+/-) and wildtype (5-HTT+/+) mice originating from heterozygous mating pairs fully backcrossed onto C57BL/6J genetic background [9]. All animals were bred and housed in the animal facility of the Center for Experimental Molecular Medicine at the University of Wuerzburg under controlled conditions (12/12 h light-dark cycle, 21±1°C room temperature and 55±5% relative humidity) and with food and water provided *ad libitum*. Mice were genotyped by PCR using genomic DNA extracted from ear tissue.

### Functional magnetic resonance imaging

**General procedure.** FMRI experiments were performed at the Department of Experimental Physics 5 (University of Wuerzburg) in cooperation with the Fraunhofer Development Center X-Ray Technology EZRT, Department of Magnetic Resonance and X-Ray Imaging on

**Table 1. Age, body weight and number of mice used in fMRI and c-Fos experiments.**

|  | Male | | | Female | | |
|---|---|---|---|---|---|---|
|  | +/+ | +/- | -/- | +/+ | +/- | -/- |
| Age [mo, mean (min-max)] | 4.9 (3.5–6.1) | 5.4 (4.0–6.1) | 5.0 (3.6–5.9) | 5.4 (5.1–5.6) | 5.4 (5.0–5.6) | 5.4 (5.2–5.6) |
| BW [g, mean±SEM] | 32.0 ± 1.3 | 33.9 ± 1.1 | 34.8 ± 1.6 | 24.4 ± 0.7[a] | 25.6 ± 0.9[a] | 29.9 ± 1.1[b] |
| fMRI [n] | 8 | 9 | 12 | 9 | 9 | 10 |
| c-Fos [n] | 4 | 4 | 6 | 2 | 2 | 2 |

BW, body weight; mo, months; g, grams; n, number of mice

a: $p<0.05$ vs.

b. Additional control mice (neutral odor): 5-HTT+/+: male (n = 1), female (n = 1); 5-HTT+/-: male (n = 1); female (n = 1); 5-HTT-/-: male (n = 1), female (n = 2).

a wide-bore ultra-high-field magnet at 17.6 T (Bruker Avance 750 WB, Bruker BioSpin GmbH, Ettlingen, Germany) with a Bruker Mini 0.75 (300 mT/m) gradient system. As displayed in Table 1 fMRI was conducted on 29 male and 28 female mice that were exposed to rat odor as an unconditional fear-evoking stimulus. A subgroup of these mice was then used to evaluate the induction of c-Fos immunoreactivity in the amygdala. In addition, some mice (male: n = 3; female: n = 4) were exposed to a neutral odor and thus served as a negative control of c-Fos induction by rat odor.

Animals were transported to the location with the 17.6 T scanner 24 h prior to the experiment to allow adaptation to the new environment to reduce stress.

All fMRI measurements were performed during the light phase between 9 AM and 7 PM. Males and females were tested independently, and the testing order was continuously alternated between the three genotypes to prevent potential time-of-day effects. Moreover, to control for sex hormone fluctuations in female mice, the individual estrous cycle stage was determined cytologically as described by McLean and coworkers directly before fMRI [58].

**Odor preparation.** Rat odor administered during fMRI was prepared from a 50 ml tube filled with male rat soiled bedding moistened with water and stored at -20˚C to prevent loss of odorous substance. The frozen bedding was slowly defrosted right before initiation of the measurement. Neutral control odor was prepared the same way, but with unused bedding material.

**Animal preparation.** Mice were pre-anesthetized in their home cages with an induction level of 0.8% isoflurane (Forene®, Abbott, Switzerland) in compressed air (1.0 l/min) to prevent pre-experimental stress through slow sleep induction. The isoflurane concentration was then elevated to 4.0% by volume for up to 10 min to ensure deep anesthesia, important for subsequent fixation of the mice in supine position in a custom-made holder. Mice were fixed at four locations to prevent head movements in the scanner: Both ears with cotton buds, back of the head on a locating surface and teeth with a strand loop. The holder was positioned in a 38 mm diameter birdcage coil, which was used for radio-frequency transmission and signal reception. The percentage of isoflurane was decreased to 1.0% to maintain lower anesthesia during the experiment.

**Vital sign monitoring.** Respiration and heart rate were monitored throughout the experiment using an air-balloon positioned ventrally underneath the mouse thorax. The active temperature control of the MRI gradient system was used to maintain a constant environmental temperature of 37˚C, which was monitored by a thermal sensor in the gradient system.

**Experimental procedure.** Structural and functional images of individual mice were registered onto a reference image selected from the measurements on the criteria of similarity to the image of the Allen Brain Mouse Atlas of the same sectional plane. This was done to

compensate for anatomical- and geometrical-dependent differences between the subjects and to use one uniform regions of interest (ROI) layout in order to eliminate the error of individual ROI drafting.

The experimental procedure consisted of eight 10 min perfusion measurement blocks each comprising of 10 labeled and 10 unlabeled MRI acquisitions. As displayed in Fig 1, three 10 min blocks were acquired during the resting state (RS), two blocks during the stimulation state (SS) and three blocks during the post-stimulation state (PS). To prevent habituation, rat (or neutral) odor was applied automatically every two seconds as an air blast of one second duration by adding it to the constant stream of air/isoflurane-mixture during SS. Two hours after SS onset, mice were sacrificed, according to IACUC standards, by cervical dislocation following a deep anesthesia with isoflurane. Brains were then immediately dissected and processed for subsequent c-Fos immunohistochemical analyses.

**Perfusion MRI with continuous arterial spin labeling.** CASL measurements were performed with a modified single coil method using a turbo spin-echo imaging sequence [59–61] already described as applicable for perfusion measurements in animal models of ischemic stroke [62]. CASL parameters are described in detail by Pham and coworkers [63]. An in-house built transmit/receive linear birdcage resonator coil with an inner diameter of 38 mm was used.

Rostral to caudal adjustment of the image plane for perfusion measurements was accomplished with coronal orientated scout images [Rapid Acquisition with Refocused Echoes (RARE), echo train length (ETL) = 20, effective echo time TEeff = 9.43 ms, repetition time TR = 7.5 s, slice thickness = 1.0 mm, number of slices = 3, field of view (FOV) = 4.0×4.0 cm, matrix of 180×180 voxels] for each animal to assure that most parts of the amygdala were within the measuring plane. The aim was to adjust the head of the mouse in a position so that the scout image was at the best rate equivalent to Bregma level -1.22 mm [64].

A radio frequency (RF) pulse located 1.4 cm away from the imaging plane based on the tagging-gradient was applied for the adiabatic inversion of arterial blood in the neck. Compensation of magnetization transfer effects was achieved by reversing the tagging-gradient before acquisition of the corresponding non-inverted control image.

RARE parameters were: ETL = 8, $TE_{eff}$ = 28.88 msec, TR = 1.0 sec, slice thickness = 1.5 mm, FOV = 1.68×1.68 cm with a matrix of 64×64 voxels. Alternating acquisition of

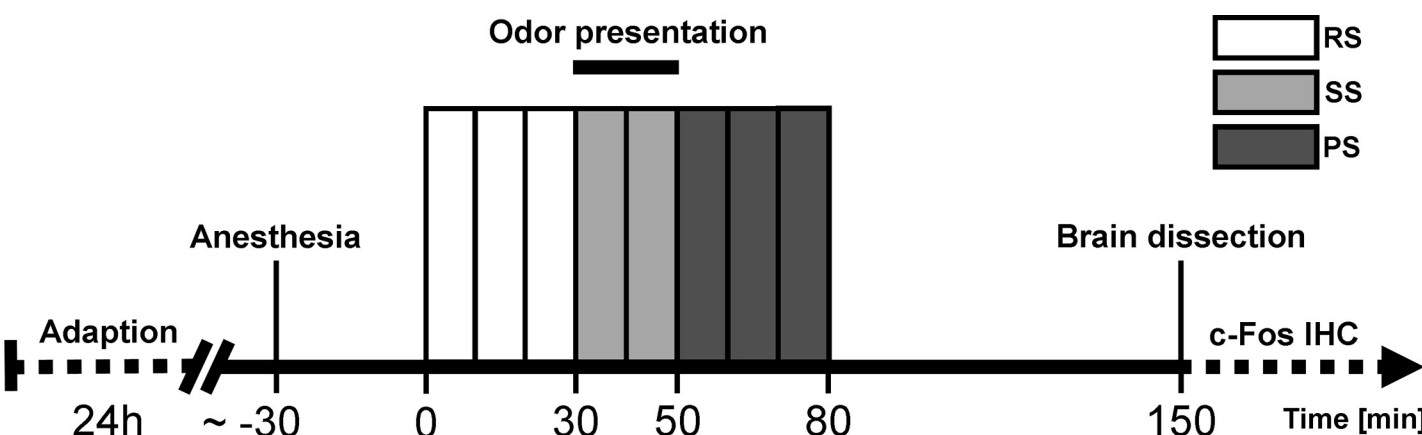

**Fig 1. Time course of the fMRI experiment.** Mice were allowed to habituate to the new environment in the adaption phase 24 h prior to the fMRI measurements. Resting state (RS, 30 min), stimulation state (SS, 20 min) and post-stimulation state (PS, 30 min) measurements comprising of separate 10 min blocks were performed on anesthetized mice. Each block consisted of 10 labeled/unlabeled MRI acquisition pairs. Two hours (120 min) after initiation of aversive or neutral odor presentation brains were dissected and processed for subsequent immunohistochemical stainings with the c-Fos antibody (c-Fos IHC).

measurements with the blood being labeled and unlabeled was repeated 10 times, resulting in a block measurement time of 10 min. All acquisitions were post-processed using MATLAB® (The Mathworks Inc., Natick, MA, USA). Images were screened for artifacts e.g. reflecting mouse head movements to prevent distorted perfusion measurements. The block-wise averaged labeled and unlabeled images were subtracted from each other and the difference representing the perfusion was then divided by the unlabeled image to finally obtain the perfusion level. The signal intensity of the unlabeled images is proportional to the amount of protons (e.g. blood) which is influenced by all kind of effects that target blood and water content as well as influences such as receiver amplification and field homogeneity which may differ for each measurement. The perfusion level indicates the proportion of perfusion relative to the overall signal intensity of each voxel, which represents a much less intereference-prone parameter. This approach extensively excludes any preexisting and constant conditions like subject specific blood supply, which might influence the measurements.

These block-wise maps were averaged again for each state to create one map per state. In order to eliminate differences in brain geometry, an anatomic image of each mouse underwent an image registration with the algorithm of Periaswamy and Farid [65,66]. The individual registration vector map was then applied to all corresponding images and maps in order to use one ROI layout for all mice to minimize inter-individual errors.

Both, percental perfusion level changes from RS to SS and SS to PS were calculated for each animal using following calculation: SS/RS x 100 or PS/SS x 100.

**Image analysis.** ROIs, i.e. total brain and the amygdaloid region comprising La, BL and Ce (termed as "amygdala" in the following), were defined in the consolidated image of all registered mouse brains as indicated in the mouse brain atlas of Franklin and Paxinos at Bregma level -0.94 mm to -1.82 mm (see Fig 2) [64].

## Quantitative immunohistochemistry for the detection of c-Fos protein

**Tissue preparation.** Brains were fixed in 4% PFA (dissolved in 1xPBS, pH 7.5) for 48 h. After treatment with sucrose brains were frozen in dry ice cooled isopentane and stored at -80˚C. Finally, they were cut into coronal sections of 50 μm thickness, subsequently split into 6 series (each consisting of up to 7 slices with amygdala), and preserved in a cryoprotectant solution at -20˚C for later use.

**c-Fos immunohistochemistry.** c-Fos immunohistochemistry was performed using the polyclonal antibody against c-Fos (made in rabbit, Santa Cruz, 1:8000; sc-52; unfortunately, this antibody has been discontinued in the meantime), applying the ABC method and 3,3´-diaminobenzidin as the substrate of the peroxidase according to the procedure described by [18]. No-Primary-Controls with omitting primary antibody incubation were performed and always resulted in the absence of any staining. In addition, positive-tissue-controls had been performed to verify the specifity of c-Fos immunoreactivity at the sub-cellular, cellular, and regional level.

**Quantification of c-Fos-immunoreactive cells.** For quantitative estimation of c-Fos-ir cells the Stereo Investigator 11 software was used in combination with a Neurolucida Microscope system from MBF Biosciences (Williston, USA). The experimenter blind for the *5-Htt* genotypes individually traced the area of La, BL, and Ce, counted c-Fos-immune-positive cells, and calculated the relative cell density (cells per $μm^2$). Finally, c-Fos-ir cell densities of every section were averaged for each animal.

**Data analysis.** All data were analyzed using SPSS Statistics version 23 (IBM Corporation, New York, USA). GraphPad Prism software version 6 (GraphPad Software, La Jolla, California, USA) was used for data graphing.

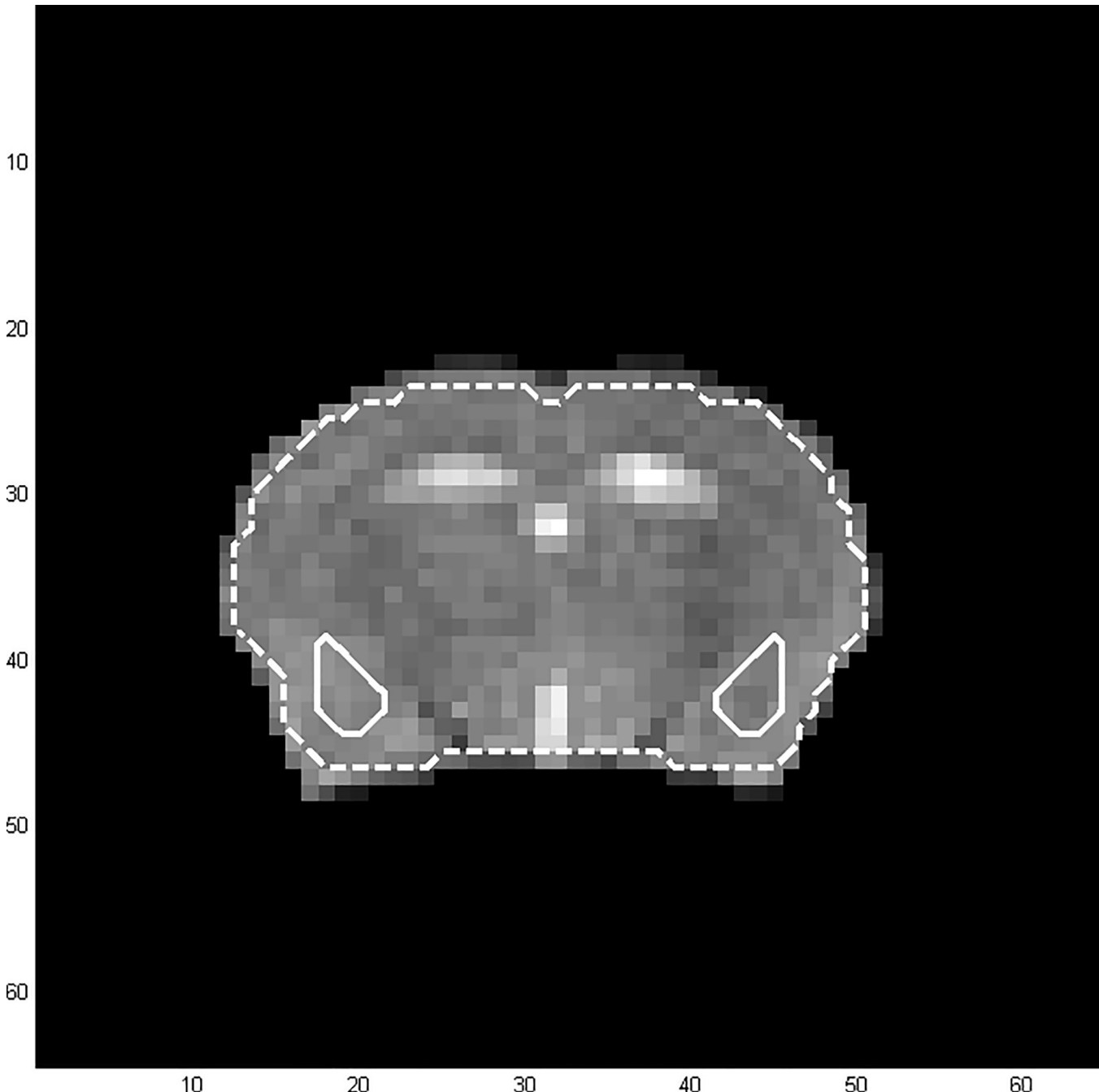

**Fig 2. Consolidated mouse brain image with defined regions of interest.** Total brain in dashed white lines and amygdala (comprising the lateral, basolateral and central amygdaloid nucleus) in solid white lines.

Comprehensive preliminary analyses were carried out to determine whether data met the assumptions of parametric tests. Data sets were visually inspected via boxplots to examine symmetry, skew, variance and outliers. Additionally, the Shapiro-Wilk test of normality and Levene's test for homogeneity of variances were performed (S1 and S3 Tables). In ANOVAs with repeated measures Mauchly's test was applied to determine

whether the assumption of sphericity has been met. Cerebral perfusion level data obtained from CASL-based fMRI scans were analyzed by two-way mixed ANOVA with *5-Htt* genotype as between-subjects factor and phase (RS, SS, PS) as within-subjects factor. In females, estrous cycle (proestrus vs. other stages) was included as an additional between-subjects factor. Relative perfusion level changes across the distinct fMRI phases were calculated as a fraction of one another (SS/RS, PS/RS, PS/SS) and analyzed by regular ANOVAs with genotype (and estrous cycle in females) as factor(s). The same statistical procedure was applied for the analysis of c-Fos-ir cell densities in the amygdala. Significant interactions were followed up with simple main effects analyses and Bonferroni *post hoc* tests for pairwise comparisons. Where applicable, Welch's ANOVA and Games-Howell post-hoc tests were used.

The Pearson correlation coefficient was performed to determine the relationship between rat odor-induced amygdala perfusion levels and c-Fos-ir cell densities in the La, BL and Ce.

The chi-square test was used to check whether the two variables estrous cycle stage and *5-Htt* genotype are statistically independent.

Data are shown as means ± standard errors of the mean (SEM) unless stated otherwise. Significance levels are indicated as $^{\#}p<0.1$, $^{*}p<0.05$, $^{**}p<0.01$ and $^{***}p<0.001$.

## Results

### Resting state perfusion and odor-induced perfusion level changes in the amygdala of male mice are influenced by *5-Htt* genotype

Statistical results are summarized in S1 Table. Two-way mixed ANOVA on local perfusion in the amygdala of males (Fig 3A) yielded a strong main effect of phase ($F(2,52) = 35.626$, $p<0.0001$) along with a significant phase x genotype interaction ($F(4,52) = 2.611$, $p = 0.046$). Simple main effects analyses showed that, on the one hand, the interaction was driven by a strong genotype effect on RS perfusion ($F(2,15.9) = 6.810$, $p = 0.007$). *Post hoc* tests revealed significantly elevated RS perfusion levels in 5-HTT+/- mice as compared to 5-HTT+/+ ($p = 0.012$; brackets with dashed lines in Fig 3A). 5-HTT-/- mice exhibited intermediate RS perfusion values that were indistinguishable from those of 5-HTT+/- and 5-HTT+/+ littermates ($p = 0.176$ and $p = 0.148$, respectively). Neither SS nor PS perfusion levels differed among *5-Htt* genotypes ($F(2,15.3) = 1.743$, $p = 0.208$ and $F(2,26) = 0.936$, $p = 0.405$, respectively). On the other hand, the interaction could be explained by a genotype-dependent difference in the magnitude of odor-induced signal change. Specifically, the increase in amygdala perfusion upon odor stimulation (RS→SS) was highest in 5-HTT+/+ ($p<0.0001$), intermediate in 5-HTT-/- ($p = 0.009$) and lowest in 5-HTT+/- ($p = 0.069$) mice. Similarly, PS perfusion level was significantly elevated above baseline (RS→PS) in 5-HTT+/+ ($p<0.0001$), 5-HTT-/- ($p<0.0001$) and 5-HTT+/- ($p = 0.023$) mice.

Accordingly, when SS and PS perfusion levels were expressed as a fraction of resting state perfusion level (SS/RS and PS/RS, respectively), ANOVA detected a highly significant genotype effect on both SS/RS ($F(2,26) = 5.848$, $p = 0.008$) and PS/RS ($F(2,26) = 7.201$, $p = 0.003$). Post-hoc tests indicated that both measures were significantly increased in 5-HTT+/+ mice relative to 5-HTT+/- ($p \leq 0.01$) and 5-HTT-/- ($p<0.05$) littermates (Fig 3B and 3C). Conversely, the PS/SS ratio did not significantly differ between genotypes ($F(2,26) = 0.112$, $p = 0.894$). Fig 3D visualizes percental signal changes from RS to SS, as individual percental signal changes were calculated for each voxel in the ROI amygdala and according to their value they were plotted in the corresponding intensity.

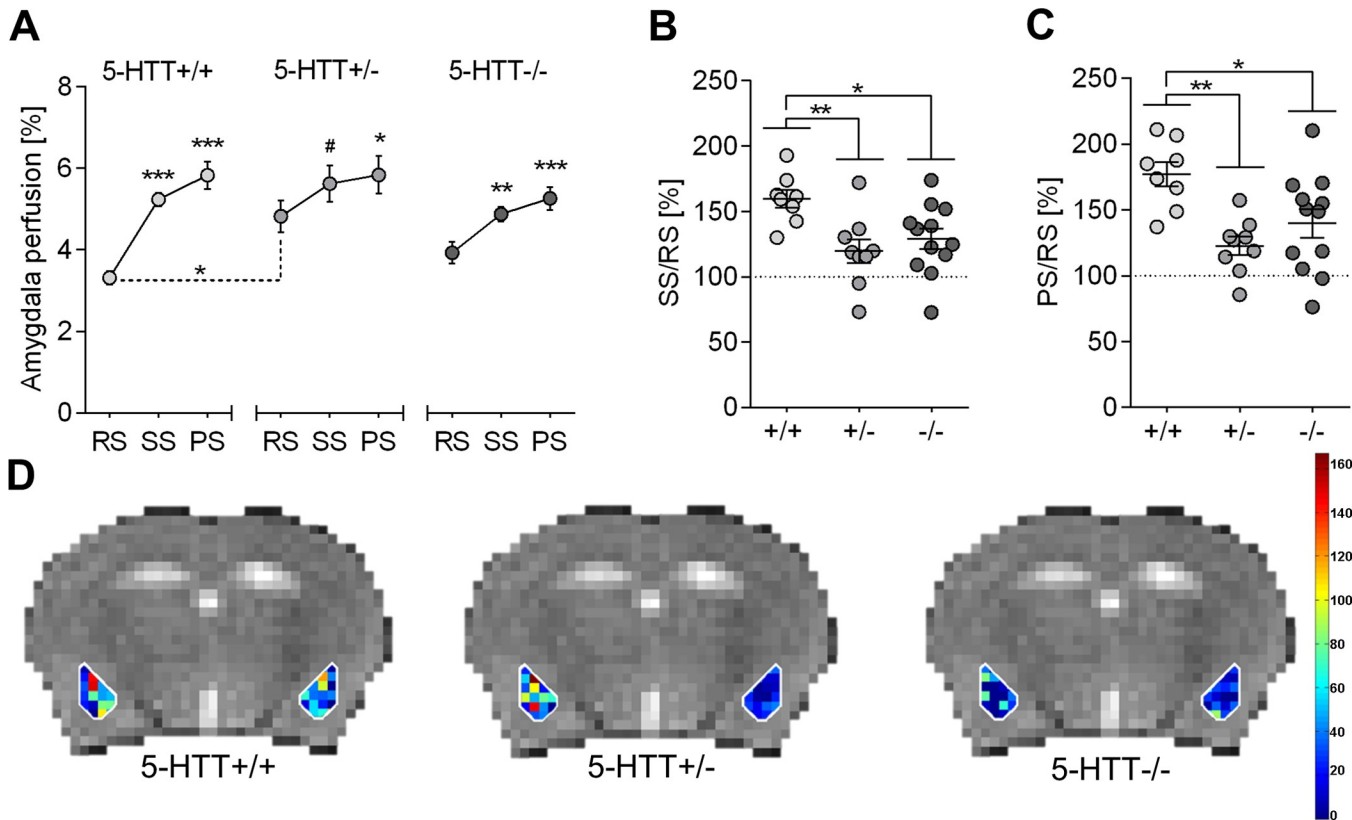

**Fig 3. CBF perfusion levels in the amygdala of male mice are influenced by the 5-Htt genotype.** A: Cerebral perfusion levels during resting state (RS), stimulation state (SS) and post-stimulation state (PS) in the amygdala of male mice of all three 5-Htt genotypes. Data represent mean perfusion level ± SEM; Statistical analysis was performed using two-way mixed ANOVA with genotype (5-Htt+/+, 5-Htt+/-, 5-Htt-/-) as between-subjects factor and phase (RS, SS, PRS) as within-subjects factor. Bonferroni post hoc tests were applied for pairwise comparisons. As with RS and SS perfusion data the assumption of homogeneity of variance is violated Welch's ANOVA was applied for testing genotype effects in these two phases followed by Games-Howell post hoc tests. Significant differences between the SS or PS state compared to the RS state within the separate genotype groups are indicated. The bracket with dashed lines point to genotype-dependent differences of RS perfusion levels. *** $p < 0.001$, ** $p \leq 0.01$, * $p \leq 0.05$, # $p \leq 0.1$. B: Data points represent individual percental increase in amygdala CBF levels in male 5-HTT+/+, +/- and -/- mice from RS to SS and mean percental increase ± SEM. C: Data points represent individual percental increase in amygdala CBF levels in male 5-HTT+/+, +/- and -/- mice from RS to PS and mean percental increase ± SEM; Statistical analysis using one-way ANOVA with genotype as between-subject factor and Bonferroni post hoc test. ** $p \leq 0.01$, * $p \leq 0.05$. D: Visual representation of perfusion level changes from RS to SS in male mice of different 5-Htt genotypes during aversive odor presentation. Individual voxel-based calculation of amygdala perfusion level changes depicted in colors corresponding to different values.

Analysis of whole-brain perfusion yielded a highly significant main effect of phase ($F(2,52) = 46.573$, $p<0.0001$), but no genotype main effect ($F(2,26) = 3.031$, $p = 0.066$) or an interaction between these two factors ($F(4,52) = 1.860$, $p = 0.131$; S1 Fig, S1 Table).

## Odor-induced c-Fos immunoreactivity in the amygdala of male mice is influenced by *5-Htt* genotype

Amygdalar cellular activation upon rat odor exposure during fMRI scans was assessed by quantitative analysis of c-Fos-ir cells in the La, BL and Ce of male mice two hours after stimulus onset (Fig 4C, 4E and 4G). We detected significant genotype effects on c-Fos-ir cell density in all three subnuclei (La: $F(2,6.09) = 6.530$, $p = 0.031$; BL: $F(2,11) = 15.594$, $p = 0.001$; Ce: $F(2,11) = 8.881$, $p = 0.005$). *Post hoc* tests indicated significantly increased c-Fos immunoreactivity in the amygdala of 5-HTT+/+ mice compared to 5-HTT+/- (La: $p = 0.053$, BL: $p = 0.003$, Ce: $p = 0.016$) and 5-HTT-/- (La: $p = 0.040$, BL: $p = 0.001$, Ce: $p = 0.007$) littermates.

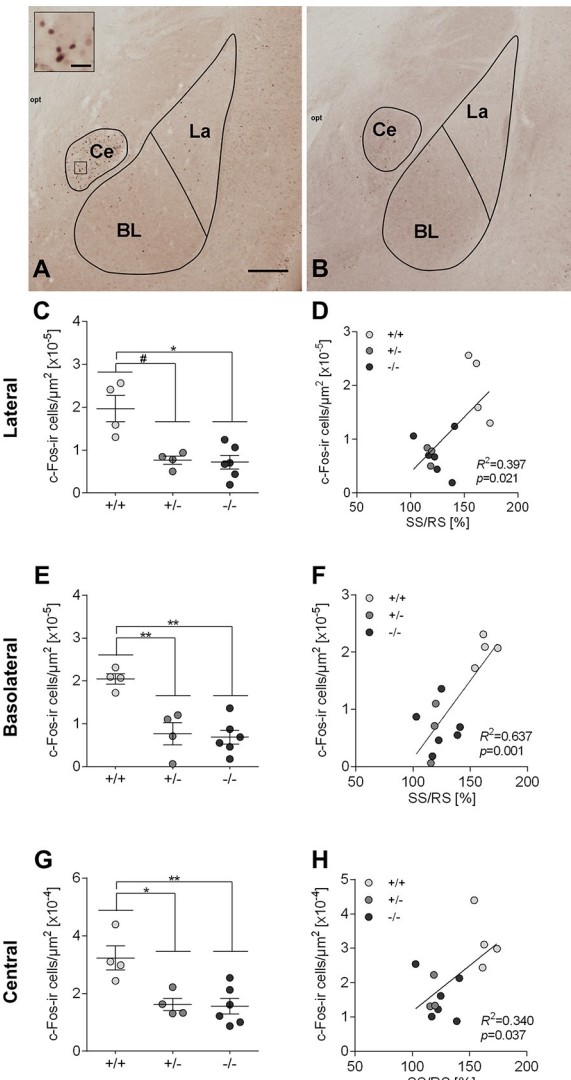

**Fig 4. The density of c-Fos-immunoreactive cells in three amygdaloid nuclei, which is influenced by *5-Htt* genotype, correlates with the percental signal change from resting state to stimulation state.** (**A,B):** Representative image of c-Fos immunostaining in a brain section of a male mouse, which received the aversive odor (**A**) or the neutral odor (**B**) during fMRI experiment. c-Fos-ir cells are discernible by means of a clear brown nucleus staining. Scale bar in A represents 250 μm for A and B, and 25 μm in the inset in A. Lateral nucleus, La; basolateral nucleus, BL; central nucleus, Ce. **(C,E,G)**: Data points represent individual and mean± SEM density of c-Fos-immunoreactive cells (number of c-Fos-ir cells per μm$^2$) in the lateral, basolateral and central amygdaloid nucleus of male mice with aversive odor exposure; Statistical analysis was performed using one-way ANOVA and Bonferroni *post hoc* tests were applied for pairwise comparisons. As in case of La data the assumption of homogeneity of variance is violated (S1 Table), Welch's ANOVA was applied and followed by Games-Howell post hoc tests. $^{**}$ p ≤ 0.01, $^{*}$ p ≤ 0.05, $^{\#}$ p ≤ 0.1. **(D,F, H):** Data points represent c-Fos-ir cell density and SS/RS. correlations are significant in the lateral (**D**) and in the basolateral (**F**), but not in the central nucleus (**H**) of the amygdala.

## Positive correlation of the density of c-Fos-immunoreactive cells and perfusion level changes points to an important relationship between perfusion and neuronal activity

Analyzing possible correlations between percental increase of perfusion from RS to SS (SS/RS perfusion level) and c-Fos-ir cell density in the amygdala resulted in a moderate, positive

correlation, which was significant for La ($r(13) = 0.630$, $p = 0.021$; Fig 4D), BL ($r(13) = 0.798$, $p = 0.001$; Fig 4F) and Ce ($r(13) = 0.583$, $p = 0.037$; Fig 4H). Moreover, PS/RS perfusion correlated significantly with c-Fos-ir cell density in all three amygdala subnuclei (La: $r(13) = 0.594$, $p = 0.032$, BL: $r(13) = 0.695$, $p = 0.008$, Ce: $r(13) = 0.587$, p = 0.035).

### Estrous cycle stage determination

The four different estrous cycle stages (proestrus, estrus, metestrus, diestrus) were fairly equally distributed across genotypes ($\chi 2(6) = 5.527$, p = 0.478; n = 28; S2 Table). Overall, however, the proestrus stage greatly predominated over the other stages ($\chi 2(3) = 20.571$, p<0.001; S2 Table). Given the significant distributional difference between estrous cycle stages (irrespective of the *5-Htt*-genotype) and the fact that proestrus female mice (with overall highest estrogen levels) are less anxious than females in the other estrous phases [67], mice were classified into two groups (proestrus vs. other stages) for subsequent analyses.

### Estrous cycle stage affects odor-induced perfusion level in the amygdala of female mice

Statistical results are summarized in S3 Table. Three-way mixed ANOVA on amygdala perfusion in female mice yielded a strong main effect of phase ($F(2,44) = 26.388$, $p<0.0001$) along with a phase x estrous cycle interaction ($F(2,44) = 5.367$, $p = 0.009$). Neither RS nor PS perfusion differed among estrous cycle groups ($F(1,22) = 0.025$, $p = 0.876$ and $F(1,22) = 0.072$, $p = 0.791$, respectively), while SS perfusion was significantly diminished in proestrus females relative to mice in other estrous stages ($F(1,22) = 6.044$, $p = 0.022$) (Fig 5A). This effect could be explained by a stronger increase in amygdala perfusion during odor exposure (RS→SS) in non-proestrus mice ($p<0.0001$) as compared to proestrus mice ($p = 0.002$). Furthermore, PS perfusion decreased significantly from SS levels (SS→PS) in non-proestrus females ($p = 0.026$) but remained elevated in proestrus mice ($p = 1$) (Fig 5A). Accordingly, when amygdala perfusion levels of the distinct fMRI phases were expressed as a fraction of one another (SS/RS, PS/RS, PS/SS), two-way ANOVA yielded the expected estrous cycle effect on SS/RS ($F(1,22) = 6.978$, $p = 0.015$; proestrus<other stages) (Fig 5C) and PS/SS ($F(1,22) = 8.358$, $p = 0.008$; proestrus>other stages) (Fig 5D) but not on PS/RS ($F(1,22) = 0.045$, $p = 0.835$). Similar results were obtained for whole-brain perfusion, though the effects were much less pronounced. Contrary to males, *5-Htt* genotype exerted no significant effects in females, neither on RS perfusion nor on rat odor-induced perfusion level changes in both ROIs (Amygdala: Fig 5B, whole brain: S2 Fig).

Finally, neither the genotype x estrous cycle interaction nor the interaction between these two factors and phase on both whole-brain and local amygdala perfusion levels were statistically significant, thereby demonstrating that the observed estrous cycle effects on stimulus-induced perfusion changes were comparable across *5-Htt* genotypes (S3 Table).

The number of c-Fos-ir cells in the amygdala of females appear to be highest in 5-HTT +/+ animals (S3 Fig). However, due to the small sample size of investigated females ($n = 6$) and the potential influence of their estrous cycle stage it is impossible to draw any valid conclusions.

## Discussion

We applied a CASL method of non-invasive ultra-high field perfusion MRI to analyse baseline perfusion levels and perfusion level changes in response to aversive odor in a coronal plane containing the amygdala, the predominant region for fear processing. We analysed female as

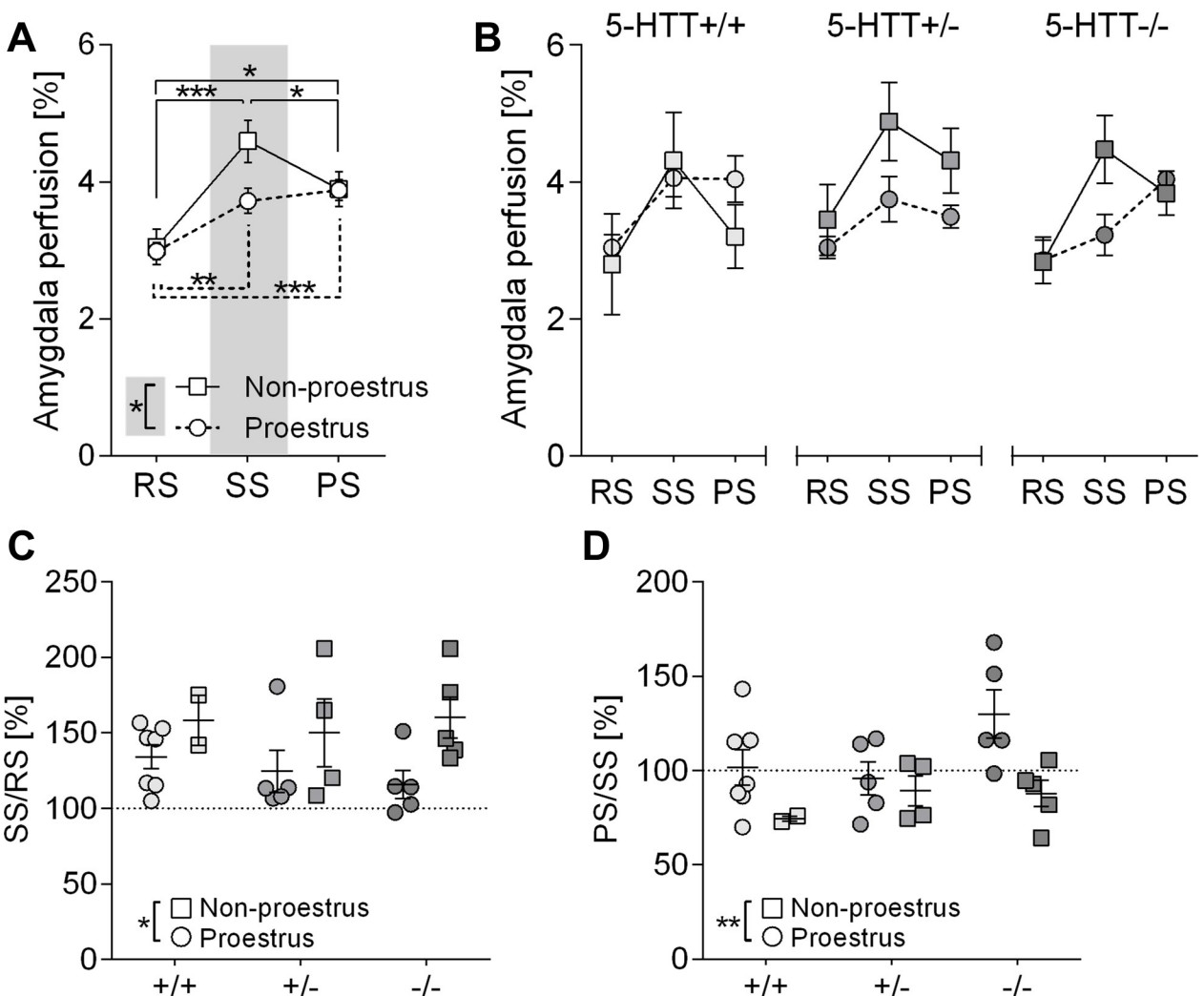

**Fig 5. CBF perfusion levels in the amygdala of female mice are affected by the estrous cycle stage but not by *5-Htt* genotype. A:** Amygdala perfusion levels during resting state (RS), stimulation state (SS) and post-stimulation state (PS) depending on the estrous stage of female mice receiving an aversive odor during SS. Data represent mean perfusion level ± SEM; Statistical analysis was performed using three-way mixed ANOVA with genotype (*5-Htt+/+*, *5-Htt+/-*, *5-Htt-/-*) and estrous cycle group (non-proestrus vs. proestrus) as between-subjects factors and phase (RS, SS, PS) as within-subjects factor. Bonferroni *post hoc* tests were used for pairwise comparisons. *** $p < 0.001$, ** $p \leq 0.01$, * $p \leq 0.05$. Grey rectangle indicates significantly diminished SS perfusion levels of mice in proestrus compared to mice in other estrous stages. **B:** Cerebral perfusion levels during RS, SS and PS in the amygdala of female mice of all three *5-Htt* genotypes depending on estrus stage. Data represent mean perfusion level ± SEM. (C, D): Data points represent individual and mean percental increase ± SEM in amygdala CBF levels in female mice depending on estrus stage from RS to SS (C) and SS to PS (D); Statistical analysis using three-way ANOVA and Bonferroni *post hoc* tests. * $p \leq 0.05$, # $p \leq 0.1$.

well as male mice of all three *5-Htt* genotypes. Subsequently performed c-Fos immunohistochemistry study served to demonstrate activation at cellular level.

Since the odor of rats is known to evoke fear in mice [35,36,38,39,68–70], and that predator odors evoke an increase in c-Fos expression levels in different amygdala nuclei (as demonstrated by increased number of c-Fos-positive cells [40–42], we assumed that a higher amygdala activation may become apparent after the presentation of the odor of rat soiled bedding as an aversive stimulus, even if the mice were under anesthetic. Previous work has shown the capability of several aversive odors including predator urine volatiles to evoke responses in the brain of anesthetized mice [71–74]. As in our study in all experimental animals amygdala and total brain perfusion levels increased from RS to SS this assumption seemed to be right.

Furthermore, as we detected more c-Fos-ir cells in the amygdala of mice exposed to the aversive rat odor than in mice exposed to neutral odor, what we show qualitatively, we conclude that rat odor is experienced as aversive by anaesthetized mice during fMRI. We assume that in association with the general perfusion increase the applied aversive rat odor represents a scent experience of special quality resulting in sustained activation of neurons. This is represented by the long-lasting high perfusion levels in male mice of all three *5-Htt* genotypes even after termination of aversive odor presentation as well as by an increased amount of c-Fos-ir cells.

An influence of *5-Htt* genotype on amygdala perfusion was revealed exclusively in male animals, with 5-HTT+/- mice exhibiting significantly higher RS perfusion levels compared to 5-HTT+/+ animals. In previous human studies also applying the CASL-method S-allele carriers, which may be best comparable to 5-HTT+/- mice with lower, but not completely absent 5-HTT protein levels, were shown to have an increased RS CBF compared to L-allele carriers (best comparable to 5-HTT+/+ mice) in the amygdala, but not in global brain [46,47]. These human studies revealed RS perfusion differences analyzing probands of both sexes. As, in contrast, Viviani and coworkers did not reveal an association between the *5-HTT*LPR polymorphism and baseline brain perfusion in a cohort of healthy individuals applying CASL [48], we think that our study is of importance.

While mice of all three *5-Htt* genotypes reached approximately the same perfusion levels during SS and PS, male 5-HTT+/+ mice showed a significantly higher percental perfusion level change from RS to SS (and PS) than 5-HTT+/- and 5-HTT-/- mice. These findings correspond to the increased density of c-Fos-ir cells in 5-HTT+/+ mice compared to 5-HTT+/- and -/- mice in all three amygdaloid nuclei. Correlation analysis provided evidence for a relationship of c-Fos-ir cell densities with perfusion level changes from RS to SS (and with a bit weaker significance from RS to PS). For a comprehensive interpretation of these correlation results one has to consider that the transcription factor c-Fos is shown to be differently regulated in individual brain regions, is discussed to be differently regulated in inhibitory and excitatory neurons, and that the time course of the induction and decay of Fos depends on the kind and strength of the stimulus itself [45,75,76]. Some brain regions seem to express no c-Fos at all despite various treatments [44]. Since the significance levels of our correlation analyses are not very different between Ce, a region with a high density of inhibitory neurons, and La, a region with a lower density of inhibitory neurons, we assume that the results of these correlation analyses do not reflect the type and composition of the amygdaloid nuclei analyzed. Therefore, there is currently no reason to doubt the validity of a positive correlation between c-Fos-positive cells and SS/RS perfusion. However, since the use of transcription inhibitors can block long-term synaptic plasticity [77], which is associated with changes in the expression of IEGs and their downstream genes, it would make sense to investigate in a future experiment whether the use of transcription inhibitors is able to reduce changes in perfusion levels from RS to SS shown in this study. This would highlight the physiological nature of c-Fos induction in relation to blood flow changes.

These lower perfusion level changes from RS to SS/PS in 5-HTT-/- and 5-HTT+/- mice than in 5-HTT+/+ controls are reminiscent of findings concerning stress-induced variations in the spine density of BLA pyramidal neurons, a morphological correlate of amygdala neuronal excitability, in male 5-HTT+/+ and 5-HTT-/- mice [78]. The fact that under basal conditions spine density in 5-HTT+/+ mice was significantly lower than in 5-HTT-/- mice, and stress experience induced spinogenesis in 5-HTT+/+ to the level of non-stressed 5-HTT-/- mice but failed to further increase spine density in 5-HTT-/- mice, can be interpreted as a "ceiling" effect. Diminished increase in amygdala perfusion upon predator odor and lack of stress-induced spinogenesis in 5-HTT-deficient mice suggest a dysfunction of neuroadaptive

mechanisms that are meant to enable coping with aversive stimuli and also supports the "tonic model" of *5-HTT*LPR function first suggested by Canli and Lesch in 2007 [5].

In contrast to the results with using males, females miss a clear effect of the *5-Htt* genotype on the level of amygdala perfusion.

Female mice were shown to be in different estrous stages during the experiment. Although the group size of particular estrous stage groups was rather small, a significant interaction between estrous stage and fMRI phase became statistically discernable in whole brain perfusion measurements, which was even stronger in amygdala measurements. Proestrus females (with high estrogen/progesterone levels) showed a lower increase in perfusion during odor exposure relative to non-proestrus mice. As several studies demonstrated that females in the high-estrogen proestrus phase exhibited low anxiety-levels [67], less fear and a stronger fear extinction [79,80], as well as reduced neural activity in the bed nucleus of stria terminalis in response to an innate fear-inducing stimulus than females in low-estrogen phases [81] we can assume that the low SS perfusion levels we detected in our proestrous-staged mice indicate low anxiety levels of these mice.

Furthermore, PS perfusion levels did not differ significantly from SS levels in male mice (irrespective of the *5-Htt* genotype). This missing reduction of CBF during PS in male mice could be the result of a long-lasting elevation of extracellular 5-HT levels after fearful experiences, reaching their maximum approximately 30–40 min after stimulus presentation with only a slow decrease to baseline levels [82,83]. Long-lasting high extracellular 5-HT levels in the amygdala could keep them in an activated status. In order to reveal a decrease of perfusion levels during PS an extension of CBF measurement during this fMRI state would be helpful.

In contrast to the results with male mice, in (low-estrogen) non-proestrus female mice PS perfusion levels decreased significantly from SS, but remained elevated like in male animals in (high-estrogen) proestrus females. Beside the variability and the influence of the estrous stage within the female group, the sex-specific processing of social cues could explain the differences between the results of both sexes. Male and female mice differently respond to the same stimuli presumably resulting in contrary behavioral output [84,85]. In general, obvious sex differences in rodents exist that influence 5-HT modulation in the amygdala, potentially derived from differential 5-HT receptor expression [86], varying 5-HT levels, extracellular [87] as well as in amygdala tissue extracts [88], both in neutral and stressful conditions. In summary, these obvious differences between males and females point to sex differences in fear processing in which the amygdala is involved.

Interestingly, the density of c-Fos-ir cells was ten times higher in the Ce compared to the La and BL in male as well as in female mice independent of their *5-Htt* genotype. The Ce, which is–in contrast to the BLA with appr. 80% glutamatergic neurons—mainly comprised of GABAergic neurons, forms connections to brain areas, including the periaqueductal grey [89,90], that mediate defensive behaviors and regulate fear responses [23]. Vice versa 5-HT fibres in the La, BL and Ce mainly orginate in dorsal raphe neurons (DRNs) [25,27,91–93], a part of the periaqueductal grey, and sparsely from median raphe neurons [26,27,29]. These 5-HT innervations originating in DRNs are known to modulate the amygdaloid microcircuitry encoding the fear response after negative stimuli. Assuming that 5-HTT+/+ mice display lower and 5-HTT-/- mice higher fear and anxiety-like behaviors, the critical role of the 5-HTT in the underlying mechanisms is undoubted.

As in all three amygaloid nuclei c-Fos-ir cell densities *post mortem* (but also percental perfusion change from RS to SS during MRI) were significantly increased in 5-HTT+/+ compared to 5-HTT+/- and 5-HTT-/- mice the question for the cell type responsible for this increased amygdala activation emerges, e.g. the ratio between c-Fos-positive glutamatergic and GABAergic cells in the BLA would be of great interest. Presumably, *5-Htt*-genotype dependent

alterations in the local inhibitory circuits with consequences on the amygdaloid output could be one approach to explain behavioral differences between mice of various *5-Htt*-genotypes [15,17,18,94].

In contrast to the results of our mouse fMRI study, it was reported that human S-allele carriers show amygdala hyperreactivity in response to the presentation of angry and fearful faces [49–55] suggesting *5-HTT*LPR-dependent differences in face processing [95–99]. Therefore, our results of exaggerated increase of CBF levels from RS to SS in 5-HTT+/+ mice, and not in 5-HTT+/- and -/- mice, seem at first to be contradictory. However, this exaggerated increase of CBF levels from RS to SS in 5-HTT+/+ mice, and not in 5-HTT+/- and -/- mice, is supported by the tonic activation model of *5-HTT*LPR function suggested by Canli and coworkers [46,51], for review: [5]), but not by the standard phasic activation model of *5-HTT*LPR function. Tonic firing of 5-HT neuron population activity seems related to the extra-synaptic tonic 5-HT levels and phasic firing activity to the rapid, high-amplitude, and intra-synaptic 5-HT release.The tonic model assumes lower RS baseline perfusion levels for LL-allele carriers relative to S-allele carriers. Perfusion imaging data obtained by Canli and coworkers exactly confirmed this assumption of elevated baseline amygdala activation levels of *5-HTT*LPR S-allele carriers compared with LL-allele carriers [46], and fit to the results of this study. However, this does not preclude the amygdala of S-allele carriers from reacting to stress with a strong phasic response on top of its elevated baseline levels [5]. Moreover, a ceiling effect is suggested as the maximal activation level observed in the amygdala during the SS condition is not different between mice of all three *5-Htt* genotypes. Therefore, it is necessary that the change from RS to that ceiling (SS) level is much larger for 5-HTT+/+ than for 5-HTT+/- mice, as they start from a lower resting baseline.

Beyond that, several differences of the human and rodent study designs, e.g. different natures of stimuli and different techniques and ways of analysis, have to be taken into account, which renders direct comparisons difficult. Different natures of stimuli are processed through different networks including the amygdala, and the role of the amygdala seems to be different in either case. During face processing the amygdala is triggering gaze changes towards diagnostically relevant facial features, which is shown to be modulated by the *5-HTT*LPR [97]. In contrast, in our study the role of the amygdala is in the processing of the threatening odor of a predator. Regarding different methodologies, BOLD-contrast imaging in human studies enables a time-related resolution of seconds [100]. In our study we applied a CASL method [57] similar to the ASL method Canli and coworkers applied in her human study from 2006 [46]. Based on our setup and parameters, the particular measurements for RS, SS and PS lasted 20–30 min, with the scope of long-term perfusion levels and long-term effects of stimulations.

To our knowledge, we were the first to show a correlation between the signal change of perfusion levels in a brain region and the density of c-Fos-ir cells in the same region *post mortem*. This suggests that the perfusion level increase in a distinct brain region derives from an accumulation of cell populations being activated, supporting the use of c-Fos-detection as a marker for neuronal activity. Moreover, the present study contributes to the findings that the *5-Htt/5-HTT* genotype modulates fear and anxiety-like behaviors after aversive stimuli by means of differential activation of the amygdala, at least in male mice. Furthermore, we were the first demonstrating sex differences in amygdala perfusion levels after the exposure to an aversive rat odor, and that estrous levels seem to have a tremendous influence on the activity of the amygdala. Last but not least, we provide an important contribution to the controversial discussion on the topic "phasic and/or tonic model of *5-HTT*LPR function" in the human imaging literature. Nevertheless, our results support the notion that fMRI investigations are an appropriate tool to study brain activation patterns in the field of fear and anxiety research, but also in other research fields.

## Supporting information

**S1 Fig. Cerebral perfusion levels in whole brain of male mice are not influenced by the *5-Htt* genotype.** Cerebral perfusion levels during resting state (RS), stimulation state (SS) and post-stimulation state (PS) in whole brain of male mice of all three *5-Htt genotypes*. Data represent mean perfusion level ± SEM.
(TIF)

**S2 Fig. Cerebral perfusion levels in whole brain of female (B) mice are not influenced by the *5-Htt* genotype.** Cerebral perfusion levels during different fMRI states in whole brain of female mice of all three *5-Htt* genotypes depending on their estrous stage receiving an aversive odor during SS. Data represent mean perfusion level ± SEM.
(TIF)

**S3 Fig. Density of c-Fos-immunoreactive cells in the investigated amygdaloid nuclei with aversive odor presentation during fMRI experiment. (A, B, C):** Individual and mean density of c-Fos-immunoreactive cells (number of c-Fos-ir cells per $\mu m^2$) in the lateral, basolateral and central amygdaloid nucleus of female mice with aversive rat odor exposure.
(TIF)

**S1 Table. Descriptive and inferential statistics of fMRI measurements with male mice of different *5-Htt* genotypes.**
(DOCX)

**S2 Table. Estrous cycle staging in female mice.**
(DOCX)

**S3 Table. Descriptive and inferential statistics of fMRI measurements with female mice of different *5-Htt* genotypes.**
(DOCX)

## Acknowledgments

Special thanks go to Sabine Voll, Marion Winnig, and Eva-Kristin Broschk for excellent technical assistance.

## Author Contributions

**Conceptualization:** Jonas Waider, Peter M. Jakob, Klaus-Peter Lesch, Angelika Schmitt-Böhrer.

**Data curation:** Jann F. Kolter, Markus F. Hildenbrand, Sandy Popp, Stephan Nauroth, Jonas Waider, Angelika Schmitt-Böhrer.

**Formal analysis:** Jann F. Kolter, Sandy Popp, Jonas Waider.

**Funding acquisition:** Klaus-Peter Lesch, Angelika Schmitt-Böhrer.

**Investigation:** Jann F. Kolter, Markus F. Hildenbrand, Stephan Nauroth, Julian Bankmann, Lisa Rother.

**Methodology:** Jann F. Kolter, Markus F. Hildenbrand, Stephan Nauroth.

**Project administration:** Markus F. Hildenbrand, Klaus-Peter Lesch, Angelika Schmitt-Böhrer.

**Resources:** Jürgen Deckert, Peter M. Jakob, Klaus-Peter Lesch.

**Software:** Sandy Popp.

**Supervision:** Markus F. Hildenbrand, Jonas Waider, Jürgen Deckert, Esther Asan, Peter M. Jakob, Klaus-Peter Lesch, Angelika Schmitt-Böhrer.

**Validation:** Markus F. Hildenbrand, Jonas Waider, Angelika Schmitt-Böhrer.

**Visualization:** Jann F. Kolter, Sandy Popp, Angelika Schmitt-Böhrer.

**Writing – original draft:** Jann F. Kolter, Markus F. Hildenbrand, Sandy Popp, Esther Asan, Angelika Schmitt-Böhrer.

**Writing – review & editing:** Jann F. Kolter, Sandy Popp, Angelika Schmitt-Böhrer.

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
