## [Decision Letter · Decision Letter 0]

28 Oct 2020

PONE-D-20-29044

Serotonin transporter genotype modulates amygdala resting state perfusion and amygdala reactivity to aversive stimuli depening on sex and estrous cycle stage

PLOS ONE

Dear Dr. Schmitt-Boehrer,

Thank you for submitting your manuscript to PLOS ONE. After careful consideration, we feel that it has merit but does not fully meet PLOS ONE’s publication criteria as it currently stands. Therefore, we invite you to submit a revised version of the manuscript that addresses the points raised during the review process.

We look forward to receiving your revised manuscript.

Kind regards,

Tamas Kozicz

Academic Editor

PLOS ONE

Journal Requirements:

2.  To comply with PLOS ONE submissions requirements, please provide methods of sacrifice in the Methods section of your manuscript.

3.  Thank you for including your ethics statement:

"The present work complies with current regulations regarding animal experimentation in Germany and the EU (European Communities Council Directive 86/609/EEC). All experiments were approved by the local authority and supported by the ‘Animal Welfare Officer’ of the University of Wuerzburg (reference number 55.2-2531.01-81/10).".

i) Please amend your current ethics statement to confirm that your named ethics committee specifically approved this study.

For additional information about PLOS ONE submissions requirements for ethics oversight of animal work, please refer to http://journals.plos.org/plosone/s/submission-guidelines#loc-animal-research  

ii) Once you have amended this/these statement(s) in the Methods section of the manuscript, please add the same text to the “Ethics Statement” field of the submission form (via “Edit Submission”).

Additional Editor Comments (if provided):

Your manuscript has been evaluated by two expert reviewers. They both have raised several major concerns. Specifically they raised serious concerns about study design and statistics.

Please address the reviewers' comments and critique in your revised manuscript and include a point-by-point rebuttal letter with your revised submission.

Reviewers' comments:

Reviewer's Responses to Questions

**Comments to the Author**

1. Is the manuscript technically sound, and do the data support the conclusions?

Reviewer #1: Partly

Reviewer #2: Yes

2. Has the statistical analysis been performed appropriately and rigorously? 

Reviewer #1: No

Reviewer #2: Yes

3. Have the authors made all data underlying the findings in their manuscript fully available?

Reviewer #1: Yes

Reviewer #2: Yes

4. Is the manuscript presented in an intelligible fashion and written in standard English?

Reviewer #1: Yes

Reviewer #2: Yes

5. Review Comments to the Author

Reviewer #1: Angelika G. Schmitt-Boehrer and co-workers combined fMRI and cFos immunohistochemistry to assess the activity in the amygdala upon predator odor exposure in mice. Authors used both female and male wildtype, serotonin transporter knockout and heterozygous mice. The idea to find a correlation between cFos and fMRI signal is interesting. The question, how the cycle of female sex hormones affects the amygdala activity and if this can be detected with these tools is also an important question. There are positive findings which are interesting enough to be published in future, but this reviewer found some major critical points and some minor issues which have to be addressed in order to increase the scientific impact of the manuscript.

Major remarks:

1. Experimental design:

Authors applied 4-6 male mice per genotype for the cFos labeling. This is relatively low, but potentially still acceptable sample size in morphology. Authors included female mice also, but here they processed only two (!) mice for histology. Considering, that the female amygdala is strongly affected by the estrus cycle-related hormonal changes, this sample size does not allow to evaluate the differences related to the hormonal changes accurately. Authors state this limitation multiple times in their manuscript, but stating this problem does not solve it. Because of this, the statement "estrous levels seem to have a tremendous influence on the activity of the amygdala" is not supported by the current work satisfactorily.

2. Statistics:

This issue is strongly linked to the design question (see at remark 1), and to the sample size problem.

A) ANOVA requires normal distribution of data, and homogeneity of variance. Authors are asked to show the statistical tests which approve that the datasets analyzed here, pass these criteria.

B) Also, a reliable power analysis is required here, which supports that one (!) mouse is enough for a neutral odor control. C) Still to statistics, correlation analyses also require a minimal sample size. How was the minimal sample size determined here? Do the datasets meet these criteria?

3. Contradictory statement in the results

Authors say that "estrous cycle stages (proestrus, estrus, metestrus, diestrus) were fairly equally distributed across genotypes" and then they express that "significant distributional difference between estrous cycle stages" occurred, which is contradictory.

4. Authors marked the central amygdala in the histological images, but they encircled the stria terminalis, and a part of the intramygdaloid division of the bed nucleus of the stria terminalis also. This could should be corrected.

5. Authors did not describe the controls for immunohistochemisitry. They did not specify the antibody used (Catalog No missing). If Authors used the same antiserum as used in Ref. 18, this was a well-trusted serum, but Ref. 18 does not say anything about specificity test in mice with this serum either. This should be given.

6. Critics on the concept to search for correlation between fMRI signal, and cFos immunoreactivity

Authors try to find a correlation between cFos immunorectivity and the brain tissue activity as assessed by fMRI tools. This is an exciting and interesting idea, but Authors did not state, consider and discuss an important limitation of the c-Fos mapping (for review see Kovacs KJ Journal of Neuroendocrinology 20, 665–672). One has to consider, that many types of inhibitory neurons do not express this immediate early gene, even, if they are highly active. Therefore, the activation of a particular brain region that is made up mainly by such inhibitory neurons might not be visible as an area with increased c-Fos immunoreactivity in a histological preparation. But, due to the higher metabolic activity of those inhibitory cells, the blood flow might increase. This will in inhibitory areas ultimately weaken or even abolish the correlation between fMRI signal and cFos. Consequently, if one takes into consideration that in many brain areas the proportion of inhibitory (i.e. potentially no cFos expressing) and excitatory (i.e. usually strongly cFos reactive) neurons differs, the strength of the correlation between cFos and fMRI signal will differ from brain area to brain area. Authors may discuss this issue, and check if the strength of correlation between cFos and fMRI is different for each area. It would be also interesting to see and discuss if this is somehow related to the neurochemical character of the neuron populations (and the proportion of those) found in a particular brain area.

Another aspect of the same question is that if a neuron shows electric activity (i.e. it fires), does not necessarily produce more cFos protein. (If neurons would do so, the control, in this case neutral odor group brain tissues, would be highly c-Fos positive also.) The occurrence of c-Fos is estimated in a neuron if the strength of the stimulus is above a certain set point, that induces a higher-order neuronal adaptation initiating changes at transcriptional level. Authors may discuss, if a below-set point-stimulus, that activates the neurons, but, does not induce cFos, may increase the metabolic activity of the cells leading to changes of the blood flow and increased BOLD signal. If this is possible, how does this interfere with the correlation between fMRI and cFos signal?

Minor remarks:

1. Authors may check the manuscript for typos carefully. Some examples: ln. 66. "conncected", ln. 462 "amgydala", ln. 471 "appying".

2. ln. 521. The term "gender" in this context is not the right choice. Instead, this reviewer would suggest to use the term "sex" here.

Question:

What do we know about the olfactory system in this knockout mouse strain? Do they have normal olfactory perception? This question arises since the serotonin transporter is present in the olfactory system (Sur et al 1996 Neuroscinence, pp 217-231). How do we know that -/- and +/- mice do perceive the rat odor properly?

Reviewer #2: In the present study Schmitt-Boehrer et al., used continuous arterial spin labeling (CASL) in combination with c-fos immunohistochemistry for investigating amygdaloid reactivity during baseline conditions or exposure to aversive odor in male and female serotonin transporter (5-HTT) transgenic mice. The authors demonstrated that an aversive odor is associated with blunted reactivity of the amygdala in male 5-HTT +/- and -/- mice compaired to +/+ controls, which corelates with c-fos activation of amygdaloid sub-nuclei. Furthermore, they uncovered that estrous levels in female mice directly influence amygdala perfusion levels. Though these findings are novel and not uninteresting, the manuscript has some matters that needs to be addressed.

Major comments

In the present study the neutral odor control group is of n=1 in both imaging and immunohistochemistry experiments. This is a problem for the statistical relevance of the result, and hence it cannot be used to draw any scientific conclusions. Please either remove the n=1 neutral odor group from all your graphs (i.e. figure 3, figure 4 and 5) or alternatively include more animals in the neutral odor group to reach a proper n-size equivalent to that of your test-groups.

Please include the PS/RS perfusion results in figure 3, as you describe these results in the text p. 15 line 303.

Also, did you observe any changes from RS to PS as individual percental signal changes for each voxel in amygdala? This could be an interesting observation, as you didn’t see any significant changes for RS to SS in Figure 3C.

In the figure 3 legend (p. 16, line 326-328), you mention whole-brain perfusion results, however these are not included in the figure nor described in the results-section - Please revise.

In the results-section 3.5 (p. 19-20, line 384-399) you have omitted the data from figure 5B, please revise.

Minor comments

You have misspelled a word in your title “…depening on sex and estrous cycle stage” should be “….depending on….”.

Please consider revising your title to clarify - i.e. include that these experiments were performed in mice, and shorten if possible.

In the discussion p.22, line 439-440: You are missing something in this sentence, maybe two pronouns, such as “we” and “a”.

In the discussion (p.22 line 443-444), If you want to emphasize the comparison between the neutral odor group and the aversive odor group, please add more animals to the neutral-odor group of the experiment to have sufficient statistical basis for this.

Discussion p. 22, line 448-451, you conclude that the olfactory stimulus is aversive based on amygdala reactivity, this a strong claim without the support from behavioral data such as conditioned place aversion – please revise.

Discussion p. 25-26, line 529-547, consider to leave out the circuitry-discussion as it is not relevant for your study approach, and the discussion is fairly long.

6. PLOS authors have the option to publish the peer review history of their article (what does this mean?). If published, this will include your full peer review and any attached files.

Reviewer #1: **Yes: **Balazs Gaszner

Reviewer #2: No

---

## [Author Response · Author response to Decision Letter 0]

8 Jan 2021

Thank you very much for your mail informing us about reviewers’ comments to our manuscript entitled “Serotonin transporter genotype modulates amygdala resting state perfusion and amygdala reactivity to aversive stimuli depending on sex and estrous cycle stage” and for the possibility to revise the manuscript according to the reviewers’ suggestions. In addition, we are grateful that we can improve the manuscript according to the PlosOne journal requirements. 

Modifications regarding journal requirements: 

Answer:

We adapted the heading style (e.g. removed the numbering) and ensured that throughout the whole manuscript the double-space paragraph format is utilized. We removed ZIP or Postal codes and street addresses from affiliations, and the order of affiliation components had been changed of small to large. Other small changes are related to the spelling of words – e.g. “Fig. 1” had been changed to “Fig 1”. And “Fig. S1” to “S1 Fig”. From the Acknowledgements we removed funding or competing interest information. And we removed the URL from the references. 

2. To comply with PLOS ONE submissions requirements, please provide methods of sacrifice in the Methods section of your manuscript. 

Answer: 

We provided more details about how mice had been sacrificed in the chapter Experimental procedure of the Methods section and wrote as follows: “Two hours after SS onset, mice were sacrificed, according to IACUC standards, by cervical dislocation following a deep anesthesia with isoflurane. Brains were then immediately dissected and processed for subsequent c-Fos immunohistochemical analyses.”

3. Thank you for including your ethics statement:

"The present work complies with current regulations regarding animal experimentation in Germany and the EU (European Communities Council Directive 86/609/EEC). All experiments were approved by the local authority and supported by the ‘Animal Welfare Officer’ of the University of Wuerzburg (reference number 55.2-2531.01-81/10).".

i) Please amend your current ethics statement to confirm that your named ethics committee specifically approved this study. For additional information about PLOS ONE submissions requirements for ethics oversight of animal work, please refer to http://journals.plos.org/plosone/s/submission-guidelines#loc-animal-research

ii) Once you have amended this/these statement(s) in the Methods section of the manuscript, please add the same text to the “Ethics Statement” field of the submission form (via “Edit Submission”).

Answer: 

We amended our ethics statement to “ The present work complies with current regulations regarding animal experimentation in Germany and the EU (European Communities Council Directive 86/609/EEC). All procedures and protocols have been approved by the committees on the ethics of animal experiments of the University of Würzburg and of the Government of Lower Franconia (license 55.2-2531.01-81/10). Sacrifice was performed under deep isoflurane anesthesia. All efforts were made to minimize suffering.“

Answer: 

We are sorry that we overlooked these data sharing requirements of PlosONE. We removed the phrase „data not shown“ in the manuscript and in the revised version of the manuscript all relevant data are available. We included two Supplemental figures (S1 Fig. And S2 Fig.) with cerebral perfusion levels in whole brain of male (S1 Fig.) and female (S2 Fig.) mice, and provided two supplemental tables with information about descriptive and inferential statistics of fMRI measurements and c-Fos immunohistochemistry using male mice (S1 Table) and of fMRI measurements using female mice (S3 Table).

1) On page 19/20 (pdf-file of the first submission) „Accordingly, when amygdala perfusion levels of the distinct fMRI phases were expressed as a fraction of one another (SS/RS, PS/RS, PS/SS), two-way ANOVA yielded the expected estrous cycle effect on SS/RS (F(1,24)=4.233, p=0.051; proestrus<other stages) (Fig. 5C) and PS/SS (F(1,24)=7.494, p=0.011; proestrus>other stages) (Fig. 5 D) but not on PS/RS (F(1,24)=0.076, p=0.786) (not shown). A less strong effect in the same direction was observed for whole-brain perfusion in female mice (data not shown). Contrary to males, 5-Htt genotype exerted no significant effects in females, neither on RS perfusion nor on odor-induced perfusion level changes in both ROIs (whole brain: data not shown; Amygdala: Fig. 5B).” was changed to “Accordingly, when amygdala perfusion levels of the distinct fMRI phases were expressed as a fraction of one another (SS/RS, PS/RS, PS/SS), two-way ANOVA yielded the expected estrous cycle effect on SS/RS (F(1,22)=6.978, p=0.015; proestrus<other stages) (Fig 5 C) and PS/SS (F(1,22)=8.358, p=0.008; proestrus>other stages) (Fig 5 D) but not on PS/RS (F(1,22)=0.045, p=0.835). Similar results were obtained for whole-brain perfusion, though the effects were much less pronounced. Contrary to males, 5-Htt genotype exerted no significant effects in females, neither on RS perfusion nor on rat odor-induced perfusion level changes in both ROIs (Amygdala: Fig 5B, whole brain: S2 Fig; S3 Table). 

2) Moreover, we included the S1 Figure displaying perfusion level changes of whole brain in male mice as well as information about descriptive and inferential statistics of fMRI measurements and c-Fos immunohistochemistry using male mice in the tables of S1 Table.

 (On page 16 (pdf-file of the first submission): “In contrast to the amygdala, three-way mixed ANOVA on whole-brain perfusion levels in males detected a highly significant main effect of phase (F(2,56)=17.790, p<0.0001), but no phase x stimulus type (=aversive rat vs. neutral odor) or phase x genotype interaction.” was changed to “Analysis of whole-brain perfusion yielded a highly significant main effect of phase (F(2,52)=46.573, p<0.0001), but no genotype main effect (F(2,26)=3.031, p=0.066) or an interaction between these two factors (F(4,52)=1.860, p=0.131) (S1 Fig, S1 Table).

Answers to the Reviewers: 

We are very glad that both reviewers appear to have a favourable impression of our work and consider the results important. We are also very grateful to the reviewers for their thorough analyses of our data, documentation and interpretation and have carefully considered their helpful suggestions. Accordingly, we have tried to adequately address all points of criticism in a revision of our manuscript. 

Answers to reviewer 1:

Reviewer: 

1. Experimental design: Authors applied 4-6 male mice per genotype for the cFos labeling. This is relatively low, but potentially still acceptable sample size in morphology. Authors included female mice also, but here they processed only two (!) mice for histology. Considering, that the female amygdala is strongly affected by the estrus cycle-related hormonal changes, this sample size does not allow to evaluate the differences related to the hormonal changes accurately. Authors state this limitation multiple times in their manuscript, but stating this problem does not solve it. Because of this, the statement "estrous levels seem to have a tremendous influence on the activity of the amygdala" is not supported by the current work satisfactorily.

Answer:

We thank the reviewer for this helpful criticism. We decided to completely remove chapter 3.6 of the original submission (page 21 of PONE-S-20-35131_15.09.2020.pdf) with c-Fos in females. 

We only mentioned at the end of the overall results part that „The number of c-Fos-ir cells in the amygdala of females appear to be highest in 5-HTT +/+ animals (S3 Fig.). However, due to the small sample size of investigated females (n=6) and the potential influence of their estrous cycle stage it is impossible to draw any valid conclusions.“

With regard to females, we will now focus on showing the results of fMRI perfusion. A group size of n = 9-10 females was available for fMRI experiments.

Reviewer: 

2. Statistics: This issue is strongly linked to the design question (see at remark 1), and to the sample size problem.

A) ANOVA requires normal distribution of data, and homogeneity of variance. Authors are asked to show the statistical tests which approve that the datasets analyzed here, pass these criteria. B) Also, a reliable power analysis is required here, which supports that one (!) mouse is enough for a neutral odor control. C) Still to statistics, correlation analyses also require a minimal sample size. How was the minimal sample size determined here? Do the datasets meet these criteria? 

Answer: 

To A) In two supplement tables we now provide descriptive and inferential statistics of fMRI measurements and c-Fos immunohistochemistry with male mice of different 5-Htt genotypes (S1 Table) and solely of fMRI measurements with female mice of different 5-Htt genotypes (S3 Table). In these S Tables we provide, inter alia, the results of the Shapiro-Wilk test of normality and the Levene’s test for homogeneity of variances. Moreover, in ANOVAs with repeated measures Mauchly’s test was applied to determine whether the assumption of sphericity has been met. As in a few cases the homogeneity of the variance and/or the normal distribution was violated, we applied the Welch's ANOVA (which can be used even when the groups have unequal variances) followed by Games-Howell post hoc tests (instead of Bonferroni post hoc tests) in these cases. Corresponding results is provided in S1 Table and in S3 Table. In „Data analysis“ of the Material and Methods section we now provide more details of statistical tests used. 

To B) As Reviewer 2 also suggested, we removed the n=1 neutral odor group from all statistical analyses and from all our graphs. That`s why, it is not necessary to perform a power analysis anymore. To C) We had a sample size of n=13 for performing a Product moment correlation according to Pearson. The calculated statistical power with n=13, a significance level (denoted by alpha) of 0.05 and the different Pearson correlation coefficients of La, BL, and Ce (see results part on page 17) was 0.8 (La), 0.9 (BL), and 0.68 (Ce). A value of around .8 -.9 is usually recommended. 

Reviewer: 

3. Contradictory statement in the results. Authors say that "estrous cycle stages (proestrus, estrus, metestrus, diestrus) were fairly equally distributed across genotypes" and then they express that "significant distributional difference between estrous cycle stages" occurred, which is contradictory…

Answer: 

We added the term “irrespective of the 5-Htt-genotype” to the sentence in question to make clear that this statement is not contradictory. „The four different estrous cycle stages (proestrus, estrus, metestrus, diestrus) were fairly equally distributed across genotypes (χ2(6)=5.527, p=0.478; n= 28; S2 Table). Overall, however, the proestrus stage greatly predominated over the other stages (χ2(3)=20.571, p<0.001; S2 Table). Given the significant distributional difference between estrous cycle stages (irrespective of the 5-Htt-genotype) and the fact that …”

Reviewer: 

4. Authors marked the central amygdala in the histological images, but they encircled the stria terminalis, and a part of the intramygdaloid division of the bed nucleus of the stria terminalis also. This could should be corrected. 

Answer: 

Many thanks for your careful viewing of these figures. You are absolutely right, we have outlined this region too generously and therefore not properly/correctly. We have now adjusted the lines that border the central amygdala. 

Reviewer: 

5. Authors did not describe the controls for immunohistochemisitry. They did not specify the antibody used (Catalog No missing). If Authors used the same antiserum as used in Ref. 18, this was a well-trusted serum, but Ref. 18 does not say anything about specificity test in mice with this serum either. This should be given. 

Answer: 

We used the same antibody purchased from Santa Cruz Biotechnology in Karabeg et al., 2013 (Re. 18) and in Auth et al., 2018. Unfortunately, this antibody has been discontinued in the meantime. 

Regarding specificity tests, we compared obtained c-Fos staining results with using available brains of differently stressed mice at the sub-cellular, cellular, and regional level with already published c-Fos staining results. Of course, so called No-Primary-Controls with omitting primary antibody incubation were performed. 

In the Material and Methods section we provide now more information: 

1) The Catalog No. Sc-52 is added.

2) No-Primary-Controls with omitting primary antibody incubation were performed and always resulted in the absence of any staining. In addition, positive-tissue-controls had been performed to verify the specifity of c-Fos immunoreactivity at the sub-cellular, cellular, and regional level.

Reviewer: 

6. Critics on the concept to search for correlation between fMRI signal, and cFos immunoreactivity Authors try to find a correlation between cFos immunorectivity and the brain tissue activity as assessed by fMRI tools. This is an exciting and interesting idea, but Authors did not state, consider and discuss an important limitation of the c-Fos mapping (for review see Kovacs KJ Journal of Neuroendocrinology 20, 665–672). One has to consider, that many types of inhibitory neurons do not express this immediate early gene, even, if they are highly active. Therefore, the activation of a particular brain region that is made up mainly by such inhibitory neurons might not be visible as an area with increased c-Fos immunoreactivity in a histological preparation. But, due to the higher metabolic activity of those inhibitory cells, the blood flow might increase. This will in inhibitory areas ultimately weaken or even abolish the correlation between fMRI signal and cFos. Consequently, if one takes into consideration that in many brain areas the proportion of inhibitory (i.e. potentially no cFos expressing) and excitatory (i.e. usually strongly cFos reactive) neurons differs, the strength of the correlation between cFos and fMRI signal will differ from brain area to brain area. Authors may discuss this issue, and check if the strength of correlation between cFos and fMRI is different for each area. It would be also interesting to see and discuss if this is somehow related to the neurochemical character of the neuron populations (and the proportion of those) found in a particular brain area. Another aspect of the same question is that if a neuron shows electric activity (i.e. it fires), does not necessarily produce more cFos protein. (If neurons would do so, the control, in this case neutral odor group brain tissues, would be highly c-Fos positive also.) The occurrence of c-Fos is estimated in a neuron if the strength of the stimulus is above a certain set point, that induces a higher-order neuronal adaptation initiating changes at transcriptional level. Authors may discuss, if a below-set point-stimulus, that activates the neurons, but, does not induce cFos, may increase the metabolic activity of the cells leading to changes of the blood flow and increased BOLD signal. If this is possible, how does this interfere with the correlation between fMRI and cFos signal?

Answer: 

Many thanks for your comment. Looking through the literature, we can confirm some of your concerns, for example that the transcription factor c-Fos is differently regulated in different brain regions, and that the time-course of Fos induction and decay varies with different inducing stimuli (aversive stimuli as well as positive stimuli, e.g. different housing conditions – Robins et al., 2020); and that some brain regions do not seem to express Fos after any treatments already tried (Dragunov and Feull, 1989).

But we couldn`t comprehend/retrace your statement that „that many types of inhibitory neurons do not express this immediate early gene“. Staiger and coworkers (2002) wrote that „By morphological phenotyping with intracellular Lucifer Yellow injections, it was found that a large majority were probably excitatory pyramidal cells, but inhibitory interneurons were also found to contain c-Fos-immunoreactive nuclei.“ The fact that we detected in our study many more c-Fos-positive cells in the striatum-like central nucleus of the amygdala (Ce) composed mainly of GABAergic neurons than in the cortex-like basolateral amygdala (BLA) with around 80% glutamatergic principal neurons and roughly 20% GABAergic interneurons confirms this statement. This could implicate that in our study in consequence of rat odor exposure the the number of c-Fos-positive Ce inhibitory neurons is higher than in the Ce of mice exposed to the neutral odor (as shown in Fig. 4A and B). But, of course, this is not proofen, as we did not further characterize these c-Fos-ir neurons in all three investigated amygdaloid nuclei e.g. via immunofluorescence double labelings of c-Fos with marker for GABAergic and glutamatergic neurons (unfortunately, not enough tissue sections hab been available). 

Therefore, I would agree to the statement of Kovacs and coworkers (2008) that „In spite of these limitations, the use of c-Fos and other regulatory- or effector transcription factors as markers of neuronal activation will continue to be an extremely powerful technique“. But, of course, we have to be careful with our conclusions as heterogeneity of cellular activation in functionally distinct parts of the brain exist and cannot always be mapped perfectly by the detection of c-Fos. 

Regarding the strength of correlation between cFos and fMRI in all three investigated amygdaloid subnuclei: correlation was shown to be highly significant in the basolateral (BL) nucleus of the amygdala (p=0.001), and correlation was shown to be significant in the lateral (La) and central (Ce) nucleus of the amygdala (p=0.021 and p=0.037, respectively). Therefore, the different levels of significance of the correlation analyses do not reflect the type and composition of the amygdaloid nuclei analyzed, at least in our study, as La and BL have approximately 80% glutamatergic principal neurons and roughly 20% GABAergic interneurons, wheras the Ce is composed mainly of GABAergic neurons. But, it is certain that the weakest correlation could be detected in the Ce. In the discussion we pointed to possible challenges with the interpretation of c-Fos as a marker for neural activation and its use for correlation analyses.

Reviewer: 

7. Minor remarks:

I. Authors may check the manuscript for typos carefully. Some examples: ln. 66. "conncected", ln. 462 "amgydala", ln. 471 "appying". 

Answer: 

We carefully checked the manuscript for typos and corrected them. 

II. ln. 521. The term "gender" in this context is not the right choice. Instead, this reviewer would suggest to use the term "sex" here.

Answer: 

According to the reviewer`s suggestion we substituted the word “gender” by the more appropriate word „sex“.

Reviewer: Question: What do we know about the olfactory system in this knockout mouse strain? Do they have normal olfactory perception? This question arises since the serotonin transporter is present in the olfactory system (Sur et al 1996 Neuroscinence, pp 217-231). How do we know that -/- and +/- mice do perceive the rat odor properly? 

Answer: 

The 5-HTT is widely distributed in the brain and can be detected in almost all brain regions. Also the olfactory bulb is densely innervated by serotonergic fibers. From several published works dealing with odor exposure and its behavioral consequences using the 5-HTT knockout mouse line we conclude that olfactory perception seems to be normal in 5-HTT-deficient mice. This assumption is supported inter alia by the the study of Adamec and coworkers (2006), who found out that predator odor exposure (in this case: cat odor) resulted in increased anxiety-related behavior in 5-HTT-deficient mice. This change in fear-like behavior requires that these mice can smell the predator odor. 

Carlson and coworkers showed that near elimination of 5-HT from the forebrain, including the olfactory bulbs, had no detectable effect on the ability of mice to perform the olfactory go/no-go task and concluded that HT neurotransmission is not necessary for the most essential aspects of olfaction (Carlson et al., 2006). However, optogenetic activation of DRN serotonergic neurons decreases odor-evoked responses in pyramidal neurons oft he anterior piriform cortex, a region important for olfactory learning and encoding of odor identity and intensity (Wang et al., 2019). To sum up, the role of 5-HT in olfaction is still under debate and should be the target of further investigations.

In the introduction, page 4, the following sentence had been added: “A previous study shows that olfactory perception is unaltered in 5-HTT-deficient mice (22).

Answers to reviewer 2:

Reviewer: 

1. In the present study the neutral odor control group is of n=1 in both imaging and immunohistochemistry experiments. This is a problem for the statistical relevance of the result, and hence it cannot be used to draw any scientific conclusions. Please either remove the n=1 neutral odor group from all your graphs (i.e. figure 3, figure 4 and 5) or alternatively include more animals in the neutral odor group to reach a proper n-size equivalent to that of your test-groups. 

Answer: 

As it is not possible for us to add more mice to the neutral odor group, we decided to completely remove this group from perfusion level analyses and adapted all statistical data as the reviewer suggested. However, we still show an image representing the results of c-Fos immunohistochemistry using brain tissue from mice exposed to neutral odor (Fig 4B). Using this qualitative approach we could display differences in the number of c-Fos-immunoreactive cells in the amygdala of mice exposed to rat odor compared to mice exposed to neutral odor (Fig 4A vs. 4B). 

Reviewer: 

2. Please include the PS/RS perfusion results in figure 3, as you describe these results in the text p. 15 line 303. 

Answer: 

As suggested by the reviewer, we included the PS/RS perfusion results in Fig 3. Graph C in Fig 3 displays these results.

Reviewer: 

3. Also, did you observe any changes from RS to PS as individual percental signal changes for each voxel in amygdala? This could be an interesting observation, as you didn’t see any significant changes for RS to SS in Figure 3C.

Answer: 

The reviewer is absolutely right, that a visual representation of perfusion changes from RS to PS would be interesting as well. However, since statistical analyzes did not reveal major differences between SS/RS (FIG. 3B) and PS/SS (FIG. 3C), we have restricted ourselves to the representation of SS/RS. This individual voxel-based calculation of amygdala perfusion level changes depicted in colors corresponding to different values simply demonstrates different levels of perfusion level changes in mice of different 5-Htt genotypes. Moreover, the generation of such visual representations of perfusion level changes is a very complex process and very time consuming. But of course we cannot be sure whether visual representations of perfusion changes from RS to PSwould not have brought new knowledge.

Reviewer: 

4. In the figure 3 legend (p. 16, line 326-328), you mention whole-brain perfusion results, however these are not included in the figure nor described in the results-section - Please revise.

Answer: 

Thank you for reading the manuscript so carefully. This sentence was not intended to be part of the figure legend. To avoid misunderstandings we have moved this sentence in question to the front of the legend of Figure 3. We have improved the results in this sentence (adapted to the revised statistical approach), and provided information that the whole brain results are available in the supplement (S1 Fig, S1 Table).

Reviewer: 

5. In the results-section 3.5 (p. 19-20, line 384-399) you have omitted the data from figure 5B, please revise. 

Answer: 

Again, thank you for reading the manuscript so carefully. In the originally submitted version of the manuscript data from Fig 5B had been provided after the Figure legend, which, however, had been very unfavorable. We have moved the sentence in question to the front of the legend of Fig 5. 

Minor comments: 

Reviewer: 

You have misspelled a word in your title “…depening on sex and estrous cycle stage” should be “….depending on….”. 

Answer: 

We have corrected this typo. 

Reviewer: 

Please consider revising your title to clarify - i.e. include that these experiments were performed in mice, and shorten if possible.

Answer: 

We changed the title to: “Serotonin transporter genotype modulates resting state and predator stress-induced amygdala perfusion in mice in a sex-dependent manner”. 

Reviewer: 

In the discussion p.22, line 439-440: You are missing something in this sentence, maybe two pronouns, such as “we” and “a”. 

Answer: 

We changed this sentence to: “Subsequently we performed a c-Fos immunohistochemistry study to demonstrate activation at cellular level.”

Reviewer: 

In the discussion (p.22 line 443-444), If you want to emphasize the comparison between the neutral odor group and the aversive odor group, please add more animals to the neutral-odor group of the experiment to have sufficient statistical basis for this. 

Answer: 

As already mentioned above we decided to completely remove this neutral odor group as it is not possible for us to add more mice to this group, and adapted all statistical data as the reviewer suggested. 

Reviewer: 

Discussion p. 22, line 448-451, you conclude that the olfactory stimulus is aversive based on amygdala reactivity, this a strong claim without the support from behavioral data such as conditioned place aversion – please revise. 

Answer: 

The reviewer is right, that we did not show such a behavioral outcome of exposing the experimental mice to the rat odor we used as predator scent in our fMRI study. We can only provide an indirect proof of the aversive nature of the used rat odor. As it is mentioned in the introduction and in the discussion, several studies have already shown that mice being exposed to rat predator scents exhibit innate defensive behaviors including flight and freezing as well as an increase in stress hormone levels(35–39). Furthermore, predator odors were shown to evoke an increase in the immediate early gene (IEG) product c-Fos in the BL, Ce and medial nucleus in rodents(40–42). And even if we could not use these mice exposed to neutral odor for proper quantitative analyses (as the group size was much too low), we could show representative images with much more c-Fos positive cells in the amygdala of rat-odor-exposed mice than in the amygdala of neutral-odor-exposed mice on a qualitative level (see Fig. 4A and B). 

Reviewer: 

Discussion p. 25-26, line 529-547, consider to leave out the circuitry-discussion as it is not relevant for your study approach, and the discussion is fairly long. 

Answer: 

We think that taking a closer look on neurobiological mechanisms in general and on compensatory mechansims already described to exist in the brain of 5-HTT-deficient mice in particular, that could be involved in the perfusion level changes in the amygdala of mice with different 5-Htt genotypes, will help to understand possible underlying mechanisms. But you are right, that the biggest part of this circuit discussion does not aid in understanding the fMRI results. We removed a few sentences on page 23 of the revised manuscript without track changes, and on pages 28-29 of the manuscript with track changes.

We would like to again thank the reviewers and the editor for all the effort they put into studying our manuscript. We feel that the manuscript has been improved by the changes we carried out according to the reviewers’ helpful suggestions, and hope that their concerns have been answered.

---

## [Decision Letter · Decision Letter 1]

5 Feb 2021

Title: Serotonin transporter genotype modulates resting state and predator stress-induced amygdala perfusion in mice in a sex-dependent manner

PONE-D-20-29044R1

Dear Dr. Schmitt-Boehrer,

We’re pleased to inform you that your manuscript has been judged scientifically suitable for publication and will be formally accepted for publication once it meets all outstanding technical requirements.

Kind regards,

Tamas Kozicz

Academic Editor

PLOS ONE

Additional Editor Comments (optional):

Reviewers' comments:

Reviewer's Responses to Questions

**Comments to the Author**

1. If the authors have adequately addressed your comments raised in a previous round of review and you feel that this manuscript is now acceptable for publication, you may indicate that here to bypass the “Comments to the Author” section, enter your conflict of interest statement in the “Confidential to Editor” section, and submit your "Accept" recommendation.

Reviewer #1: All comments have been addressed

Reviewer #2: All comments have been addressed

2. Is the manuscript technically sound, and do the data support the conclusions?

Reviewer #1: Yes

Reviewer #2: Yes

3. Has the statistical analysis been performed appropriately and rigorously? 

Reviewer #1: Yes

Reviewer #2: Yes

4. Have the authors made all data underlying the findings in their manuscript fully available?

Reviewer #1: Yes

Reviewer #2: Yes

5. Is the manuscript presented in an intelligible fashion and written in standard English?

Reviewer #1: Yes

Reviewer #2: Yes

6. Review Comments to the Author

Reviewer #1: All concerns of this reviewer were excellently addressed. The question on olfactory function of the mouse strain used was also answered. After reading the revised version of the manuscript, no new critical remarks emerged.

Reviewer #2: The authors have addressed the reviewer questions in a satisfactory manner. The manuscript is in its current state suited for publication.

7. PLOS authors have the option to publish the peer review history of their article (what does this mean?). If published, this will include your full peer review and any attached files.

Reviewer #1: **Yes: **Balazs Gaszner, MD, PhD

Reviewer #2: No

---

## [Editor Report · Acceptance letter]

11 Feb 2021

PONE-D-20-29044R1 

Serotonin transporter genotype modulates resting state and predator stress-induced amygdala perfusion in mice in a sex-dependent manner 

Dear Dr. Schmitt-Böhrer:

I'm pleased to inform you that your manuscript has been deemed suitable for publication in PLOS ONE. Congratulations! Your manuscript is now with our production department. 

Kind regards, 

on behalf of

Dr. Tamas Kozicz 

Academic Editor

PLOS ONE